# MARCH8 inhibits influenza A virus infection by targeting viral M2 protein for ubiquitination-dependent degradation in lysosomes

Xiaoman Liu[1,7], Fengwen Xu[1,7], Lili Ren[2,3,7], Fei Zhao[1], Yu Huang[1], Liang Wei[1], Yingying Wang[2,3], Conghui Wang[2,3], Zhangling Fan[1], Shan Mei[1], Jingdong Song[4], Zhendong Zhao[1], Shan Cen[5], Chen Liang[6], Jianwei Wang [2,3✉] & Fei Guo [1✉]

The membrane-associated RING-CH (MARCH) proteins are E3 ligases that regulate the stability of various cellular membrane proteins. MARCH8 has been reported to inhibit the infection of HIV-1 and a few other viruses, thus plays an important role in host antiviral defense. However, the antiviral spectrum and the underlying mechanisms of MARCH8 are incompletely defined. Here, we demonstrate that MARCH8 profoundly inhibits influenza A virus (IAV) replication both in vitro and in mice. Mechanistically, MARCH8 suppresses IAV release through redirecting viral M2 protein from the plasma membrane to lysosomes for degradation. Specifically, MARCH8 catalyzes the K63-linked polyubiquitination of M2 at lysine residue 78 (K78). A recombinant A/Puerto Rico/8/34 virus carrying the K78R M2 protein shows greater replication and more severe pathogenicity in cells and mice. More importantly, we found that the M2 protein of the H1N1 IAV has evolved to acquire non-lysine amino acids at positions 78/79 to resist MARCH8-mediated ubiquitination and degradation. Together, our data support the important role of MARCH8 in host anti-IAV intrinsic immune defense by targeting M2, and suggest the inhibitory pressure of MARCH8 on H1N1 IAV transmission in the human population.

[1] NHC Key Laboratory of Systems Biology of Pathogens and Center for AIDS Research, Institute of Pathogen Biology, Chinese Academy of Medical Sciences & Peking Union Medical College, Beijing, China. [2] NHC Key Laboratory of Systems Biology of Pathogens and Christophe Mérieux Laboratory, Institute of Pathogen Biology, Chinese Academy of Medical Sciences & Peking Union Medical College, Beijing, China. [3] Key Laboratory of Respiratory Disease Pathogenomics, Chinese Academy of Medical Sciences & Peking Union Medical College, Beijing, China. [4] NHC Key Laboratory of Biosafety, National Institute for Viral Disease Control and Prevention, Chinese Center for Disease Control and Prevention, Beijing, China. [5] Institute of Medicinal Biotechnology, Chinese Academy of Medical Sciences & Peking Union Medical College, Beijing, China. [6] McGill University AIDS Centre, Lady Davis Institute, Jewish General Hospital, Montreal, Canada. [7] These authors contributed equally: Xiaoman Liu, Fengwen Xu, Lili Ren. ✉email: wangjw28@163.com; guofei@ipb.pumc.edu.cn

The membrane-associated RING-CH (MARCH) family proteins are E3 ligases characterized by a highly conserved N-terminal RING-CH (C4HC3 RING) finger, which control the stability, trafficking, and function of various cellular membrane proteins[1–8]. MARCH8, one of 11 members of the MARCH family, was first identified as cellular modulators of immune recognitions (c-MIR)[9–12]. It constitutively polyubiquitinates conserved lysine residue in the cytoplasmic tail of the MHC II chain. The ubiquitinated MHC II is recognized by the endosomal sorting complexes required for transport (ESCRT) and delivered to late endosomes and lysosomes for degradation[13]. Non-ubiquitinated MHC II is also endocytosed, but recycles to the plasma membrane[12]. In thymic epithelial cells, MARCH8 restricts cell surface expression of MHC II, thus modulates CD4$^+$ T cell selection[14,15].

MARCH E3 ligases have been shown to limit viral replication and attenuate viral pathogenesis. MARCH8 reduces HIV-1 replication by downregulating viral envelope glycoprotein which is blocked from incorporating into virions[16,17]. Similar antiviral functions have been reported for human MARCH1 and MARCH2[18,19]. MARCH5 interacts with hepatitis B viral X (HBX) protein, which is mainly accumulated in mitochondria, and targets it for degradation, thus alleviating HBV-mediated liver disease. Furthermore, MARCH8 has very broad antiviral activity by inhibiting Ebola virus glycoprotein, human immunodeficiency virus type 1 (HIV-1) envelope glycoprotein, and avian influenza virus H5N1 hemagglutinin maturation[20]. On the other hand, some viruses including hepatitis C virus (HCV), Dengue virus, and Zika viruses recruit MARCH8 to ubiquitinate viral proteins and promote viral replication[21]. It appears that MARCH proteins have complex interactions with different viruses.

Influenza A virus (IAV) is an enveloped, segmented, negative-strand RNA virus, belonging to the *Orthomyxoviriae* Family. As an important human viral pathogen, IAV has caused severe pandemics in history, and causes seasonal flu every year[22]. IAV entry is mediated by its hemagglutinin (HA) protein[23]. Neuraminidase (NA) and the M2 protein are also incorporated into IAV viral membrane[24,25]. NA is important for the release of IAV particles from cell surface by cleaving the sialic acid, the receptor of IAV[26,27]. M2 has a role in both the assembly of IAV and the uncoating of viral core during entry[28,29]. While a number of cellular proteins have been reported to restrict IAV infection, the effect of MARCH proteins has not been examined.

In this study, we report that MARCH8 limits IAV replication both in vitro and in vivo. Instead of targeting IAV envelope glycoprotein HA, MARCH8 catalyzes ubiquitination of M2 protein at position K78, leading to M2 degradation in lysosomes. Importantly, a recombinant A/Puerto Rico/8/34 (PR8) H1N1 IAV carrying the K78R M2 protein becomes resistant to MARCH8 and exhibits greater virulence in mice.

## Results

### MARCH8 downregulates viral M2 protein from the cell surface.
Given that MARCH8 has been shown to downregulate the expression of several viral envelope proteins, we asked whether MARCH8 also affects HA incorporation into IAV particles. The efficiency of IAV HA protein incorporation into pseudovirions was evaluated by viral transduction assay and western blot analysis. MARCH8 did not affect viral transduction efficiency of viruses carrying IAV HA/NA, or IAV HA/NA/M2 (Fig. 1a). However, MARCH8 strongly inhibited the infectivity of the reporter viruses that were pseudotyped with VSV glycoprotein, which is consistent with previous report[17] (Fig. 1a). In agreement with the infectivity data, MARCH8 profoundly reduced the level of VSV-G protein both in whole-cell lysate and in virions,

whereas the expression of IAV HA and NA proteins was not affected (Supplementary Fig. 1a and Fig. 1b). Unexpectedly, we observed a marked decrease in the level of IAV M2 protein in cells expressing MARCH8 (Fig. 1b), which was further confirmed when only M2 and MARCH8 were expressed in HEK293T cells (Fig. 1c). This inhibition was lost for the E3 ligase-null mutant of MARCH8 (W114A) (Fig. 1d), suggesting that MARCH8 reduces M2 expression through its E3 ligase function.

We next examined whether MARCH8 also decreased M2 expression during IAV infection. HEK293T cells were transfected with the MARCH8 plasmid DNA before exposure to the infection by influenza A/WSN/33 (H1N1) virus (WSN) at MOI of 2. MARCH8 decreased M2 markedly without affecting viral proteins HA, NA, NP, PB1, and PB2 (Fig. 1e). We further examined cell-surface expression of M2 by immune-staining and flow cytometry. The mean fluorescence intensity (MFI) of M2 in MARCH8-expressing cells was twofold lower than in the control cells, whereas the W114A MARCH8 mutant did not exert a significant effect (Fig. 1f, g). This observation is supported by data of confocal imaging showing that MARCH8 prevented the localization of M2 protein to the cell periphery, but did not affect NP trafficking (Fig. 1h and Supplementary Fig. 1b). We then knocked down endogenous MARCH8 in HEK293T cells with siRNA, and found that MARCH8 deficiency supported a boost in M2 protein following IAV infection, while did not affect HA, NA, NP, PB1, PB2, and M1 accumulation (Fig. 1i). Cell-surface M2 was twofold higher in MARCH8-knockdown cells than that in the control cells, as shown by the results of flow cytometry (Fig. 1j, k and Supplementary Fig. 1c). Collectively, these results demonstrate that MARCH8 downregulates viral M2 protein from the cell surface.

### MARCH8 inhibits IAV replication in respiratory epithelial cells.
We then investigated whether MARCH8 affects IAV replication in respiratory epithelial cells. A549 cell clone was generated to stably express exogenous MARCH8 (Fig. 2a). The control and MARCH8-overexpressing A549 cells were challenged with IAV. Production of IAV from the infected A549 cells was monitored over different time intervals. The results showed that MARCH8 significantly reduced progeny virus titer at each time point tested with either high MOI = 1.0 or low MOI = 0.002 of IAV (Fig. 2b, c). Next, we used siRNA to deplete the endogenous MARCH8 in A549 cells (Fig. 2d). We found that cells lacking MARCH8 allowed greater IAV replication than control cells (Fig. 2e, f). We also generated knockout cell lines using CRISPR/Cas9, which was verified by western blot and further genotyped with PCR (Fig. 2g and Supplementary Fig. 2a). IAV replication significantly increased in MARCH8-depleted A549 cells (Fig. 2h, i). The replication of IAV, as determined by flow cytometry (Fig. 2j and Supplementary Fig. 2b) and immune fluorescence staining of viral NP-positive cells (Fig. 2k), was increased in MARCH8-knockout cells than in control cells. These results suggest that endogenous MARCH8 in respiratory epithelial cells substantially inhibits IAV replication.

### MARCH8 inhibits IAV replication in vivo.
To investigate the role of MARCH8 in IAV infection in vivo, we used peptide-conjugated phosphorodiamidate morpholino oligomers (PPMOs) to deplete MARCH8 in mouse lungs and then evaluated the effect on IAV infection and viral pathogenesis[30,31]. PBS, the control (PPMO-NC), or MARCH8-targeting PPMOs (PPMO-M8) were administered intranasally in mice for 2 days, before mice were either sacrificed as the baseline or challenged with 200 PFU of PR8 virus on day 0 (Fig. 3a). Depletion of MARCH8 expression in mouse lungs was observed on day 0 (before infection)

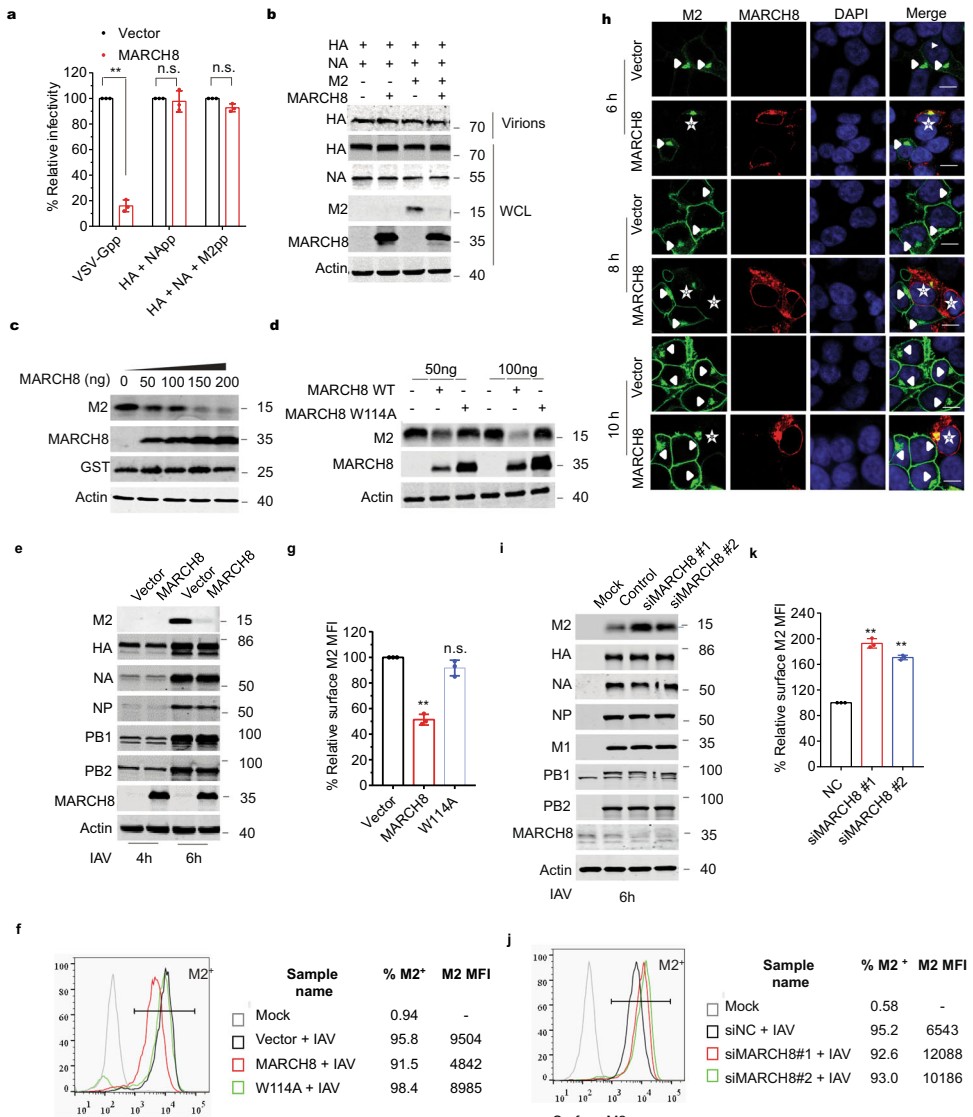

**Fig. 1 MARCH8 downregulates M2 from the cell surface. a** Infection of HEK293T cells with Lenti-GFP reporter viruses pseudotyped with VSV-G, IAV HA and NA or IAV HA, NA and M2 that were produced in control or MARCH8-expressing HEK293T cells. GFP-positive cells were scored with flow cytometry. **b** Production of Lenti-GFP reporter viruses pseudotyped with IAV HA and NA or IAV HA, NA and M2 in HEK293T cells without or with MARCH8, followed by western blot. **c** HEK293T cells were co-transfected with M2, Flag-tagged-GST, and increasing amounts of MARCH8, followed by western blot. **d** HEK293T cells were transfected with M2 and 50 or 100 ng of wild-type MARCH8 (WT) or the W114A mutant MARCH8 (W114A), followed by western blot. **e–h** HEK293T cells transfected with either a control vector or the MARCH8 were infected with WSN virus (MOI = 2). The cells were lysed for western blotting (**e**). M2 expressed on the non-permeabilized cell surface was measured by flow cytometry (**f**), and relative MFIs of surface M2 were shown in (**g**). Immunostaining of M2 and MARCH8 was shown in (**h**). M2 localization in control cells is indicated with arrows and in MARCH8-expression cells is indicated with stars. **i–k** HEK293T cells transfected with control siRNA or siRNAs against MARCH8 (siMARCH8 #1 and siMARCH8 #2) were cultured for 72 h, then infected with WSN virus (MOI = 2) for 6 h. The cells were lysed for western blotting (**i**). M2 expressed on the non-permeabilized cell surface was measured by flow cytometry (**j**), and relative MFIs of surface M2 were shown in (**k**). In **a, g**, and **k**, the values shown are normalized mean ± SD (n = 3), **P < 0.001, n.s., not significant, unpaired two-tailed Student t-test.

(Supplementary Fig. 3). Following infection, the PPMO-M8 treated mice lost more weight compared with the PBS or PPMO-NC treated mice (Fig. 3b). Concurrently, much greater titers of IAV were detected in the lungs of MARCH8-knockdown mice on day 3 and day 6 post infection, compared to PBS or PPMO-NC treated mice (Fig. 3c). Histological analysis of lung slices showed PPMO treatment itself did not induce any marked damage to the lung epithelium prior to infection (Fig. 3d, day 0). PPMO-M8 treated mice showed obvious inflammatory cell infiltration at day 3 after IAV infection and showed bronchiolitis reducing the alveolar airspace, and leukocyte infiltration at day 5

after infection. In contrast, PBS or PPMO-NC treated mice showed mild pathogenic changes at day 3 after IAV infection and the extent of bronchiolitis was markedly reduced compared with PPMO-M8 treated mice at day 5 after infection (Fig. 3d, day 3, and day 5). Overall, these results indicate an important role of MARCH8 in protecting mice from IAV infection and in mitigating IAV pathogenicity in mice.

**MARCH8 restricts IAV release**. M2 is an ion channel protein, and promotes IAV replication by modulating cellular homeostasis[32]. To

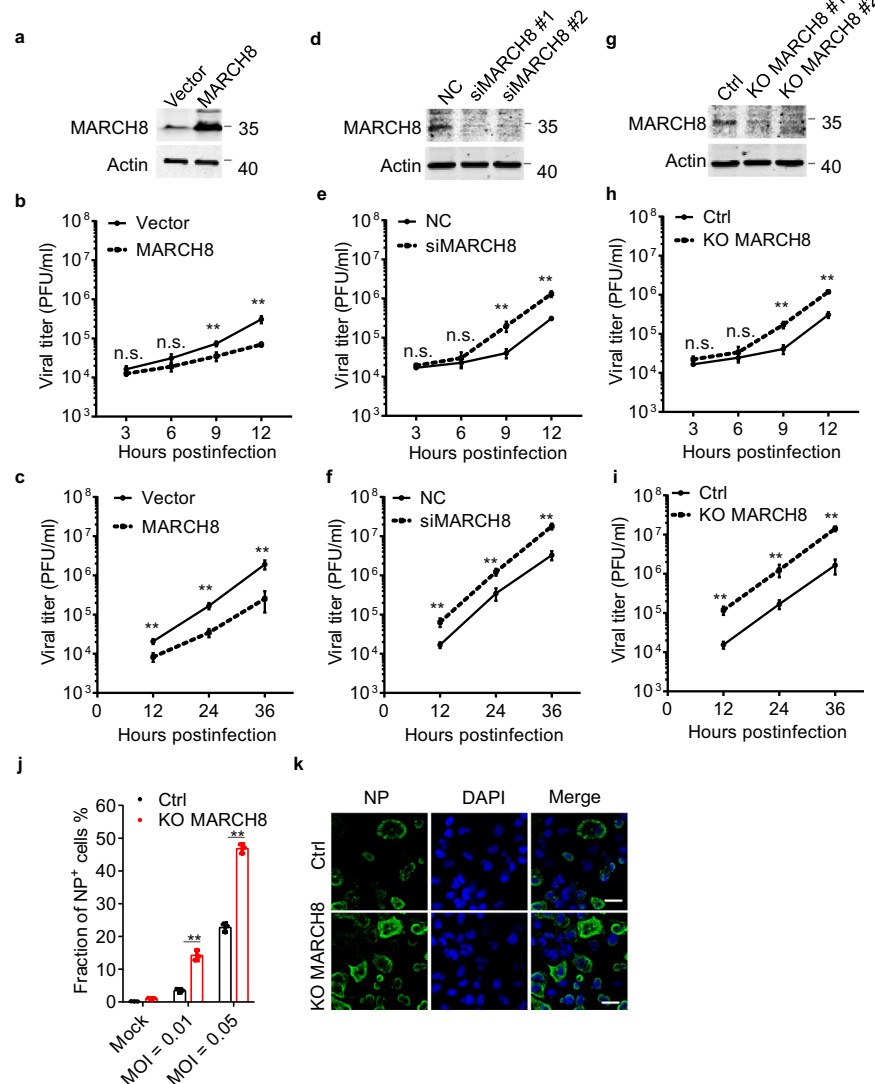

**Fig. 2 MARCH8 inhibits IAV replication in vitro. a** A549—Vector and A549—MARCH8 cells were constructed and MARCH8 expression determined by western blotting. **b**, **c** A549—Vector and A549—MARCH8 cells were infected with WSN virus for indicated times. Viral titers of supernatants were quantified by plaque assay. **d** A549 cells were transfected with siRNA targeting MARCH8. MARCH8 expression was examined by western blotting. **e**, **f** A549 cells transfected with siNC or siMARCH8 were infected with WSN virus. Viral titers of supernatants were quantified by plaque assay. **g** The endogenous *MARCH8* in A549 cells was knocked out through the use of lentiviral CRISPR-Cas9. MARCH8 expression was determined by western blotting. **h**, **i** Control (Ctrl) or *MARCH8*-knockout (KO *MARCH8*) A549 cells were infected with WSN virus. Supernatants were harvested and viral titers were detected at indicated times. **b**, **e**, **h** MOI = 1. **c**, **f**, **i** MOI = 0.002. **j**, **k** Ctrl and KO *MARCH8* A549 cells were infected with WSN virus. At 24 h post infection, viral NP protein expression was measured by flow cytometry (**j**) and confocal microscopy (**k**). Scale bar, 200 μm. Data represent averages of independent biological replicates and are presented as means ± SD (**b**, **e**, and **h**, n = 4, **c**, **f**, **i**, and **j**, n = 3). **$P < 0.001$, n.s., not significant, unpaired two-tailed Student *t*-test, without any adjustments for multiple comparisons.

dissect the role of MARCH8 in viral life cycle, we first examined whether MARCH8 affects virus binding or entry. Cells were incubated with WSN virus for 1 h at 4 °C to allow virus binding, followed by incubation with pre-warmed DMEM at 37 °C to allow virus entry. Quantitative RT-PCR data showed that MARCH8 did not affect virus binding or entry (Fig. 4a, b). We next investigated whether MARCH8 affects IAV RNA transcription and replication using a well-established minireplicon assay. The reporter pPolI-Luc produces a modified influenza virus vRNA in which the coding region is replaced with the firefly luciferase gene. Thus, the firefly luciferase activity reports the overall transcription and replication activities of the viral RNA polymerase complex. We observed similar RdRp activity in MARCH8-knockdown or -overexpressing cells compared to the control cells (Fig. 4c, d). Taken together, these data

suggest that MARCH8 does not affect the early stages of the viral replication including virus binding, virus entry, and viral RNA transcription, replication, and translation.

In IAV infection, an important function of M2 protein is to facilitate viral membrane scission, thus allowing the formation and release of progeny IAV virions[28,29,33]. We thus examined the budding of influenza viruses in HEK293T cells by transmission electron microscopy. In control cells, the budding of IAV virions could be observed at the cell surface. The MARCH8 overexpressing cells presented many IAV virions in the process of budding but impaired in membrane scission and virus particle release (Fig. 4e). Virions on the surface of MARCH8-overexpressing cells remain attached to the plasma membrane as opposed to the completely released IAV particles in the control cells, which suggests a virus

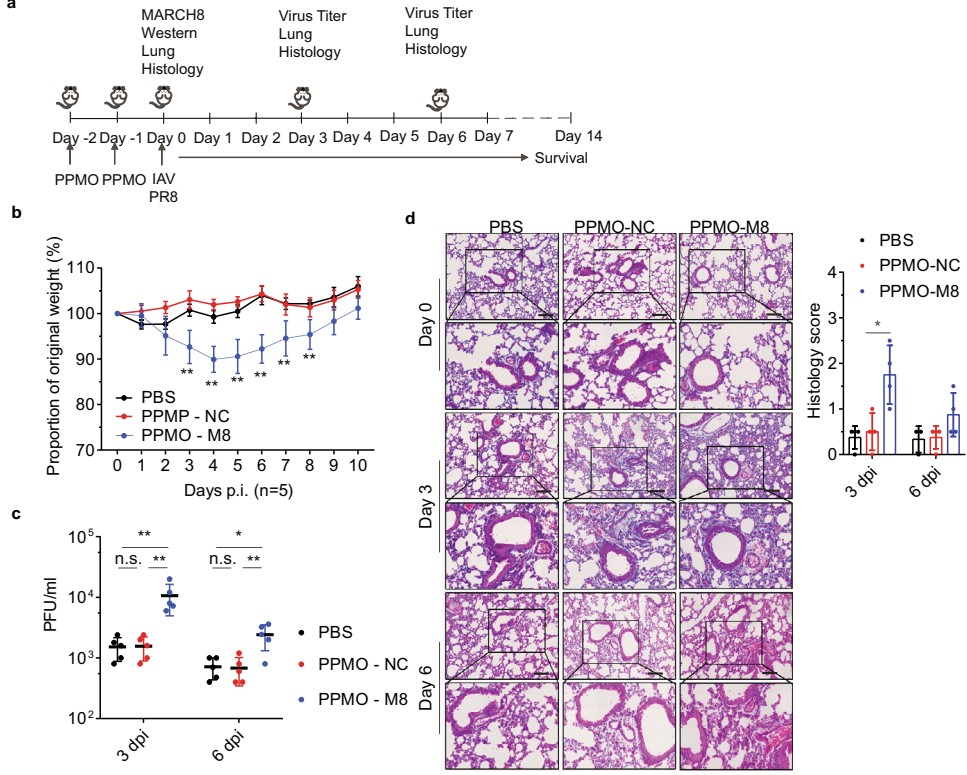

**Fig. 3 MARCH8 inhibits IAV replication in vivo.** Six-week-old male C57Bl/6 mice were administered PBS, control (PPMO-NC), or MARCH8 targeting PPMO (PPMO-M8) intranasally for 2 consecutive days. On day 0, two or three mice per group were killed before infection. Fifteen mice per group were infected with PR8 virus (200 PFU) intranasally on day 0. Five mice per group were euthanized on days 3 and 6 post infection. Bronchoalveolar lavage fluid was collected to determine virus titer, and lungs were harvested to histopathology. In five mice per group, survival was studied until day 14 post infection. **a** Diagram shows experimental plan. **b** Graph shows mouse body weight ± SD ($n = 5$ mice) up to 10 days post infection, for at least five mice per group. **c** Graph shows mean lung virus titer ±SD ($n = 5$ mice) on days 3 and 6 post infection. **d** Mouse lungs were isolated on day 0 (before infection), day 3, and day 6 post infection. H&E staining was performed on lung sections. Scale bars represent 200 µm. Histological scoring is shown as mean ± SD ($n = 4$ mice), *$P < 0.01$, **$P < 0.001$, n.s., not significant, unpaired two-tailed Student $t$-test, without any adjustments for multiple comparisons.

budding defect phenotype associated with MARCH8 expression (Fig. 4e). We further measured the level of released IAV particles in the culture supernatants by performing the IAV HA antigen-capture ELISA. Indeed, a 60% reduction in IAV production was observed for the infection of MARCH8-expressing cells (Fig. 4f). We also harvested the IAV particles by ultracentrifugation and observed a 75% decrease in IAV production by MARCH8-expression cells, as shown in Fig. 4g. We did not observe any decrease in the infectivity of pseudovirions carrying IAV HA/NA/M2 (Fig. 1a), which is likely because the viral membrane fission and virus release are carried out by HIV-1 Gag in the pseudovirus system. Together, we conclude that MARCH8 diminishes the expression of IAV M2 protein, inhibits the release and production of IAV particles.

**MARCH8 ubiquitinates the M2 protein and causes M2 degradation in lysosomes.** We then explored the mechanism by which MARCH8 decreases M2. Since MARCH8 associates with endosomes and lysosomes, and targets its substrates to lysosomes for degradation, we tested whether MARCH8 also diminishes M2 levels via the lysosomal pathway. We thus used the lysosomal inhibitors chloroquine (CQ) or bafilomycin A1 (BafA1) in the co-transfection experiments, and observed an increase in M2 expression in MARCH8-expressing cells, whereas did not observe any effect with proteasome inhibitor MG132 treatment (Fig. 5a, b). Results of immunofluorescence microscopy revealed that CQ treatment led to cytoplasmic sequestration of M2 and MARCH8 (Fig. 5c and Supplementary Fig. 4a). Using endosome marker

EEA1, lyso-tracker, and lysosome marker LAMP1, we were able to show that MARCH8 redistributed M2 from the plasma membrane to endosomes (Fig. 5d) and lysosomes (Fig. 5e, f). We also assessed the effect of MARCH8 overexpression on the intracellular distribution of M2 along with well-characterized markers of intracellular compartments, including Rab5-EGFP for early endosomes, Rab7-EGFP for late endosomes/MVB, in HeLa cells. In agreement with previous studies, co-localization of M2 with Rab5-EGFP or Rab7-EGFP was detected in MARCH8-expressing cells (Supplementary Fig. 4b, c). These data suggest that MARCH8 targets M2 to lysosomes for degradation.

To test whether MARCH8 causes M2 degradation by ubiquitinating M2, we first confirmed that MARCH8 associated with M2 by performing transient transfection and co-immunoprecipitation experiments (Fig. 5g). We next examined potential ubiquitination of M2 by MARCH8 in cells which were transiently transfected with M2 plasmid DNA or infected with IAV. We used CQ to slow down M2 degradation so that sufficient amount of potentially ubiquitinated M2 can be recovered for immunoprecipitation and western blot analysis. The results showed a significant increase in M2 ubiquitination when MARCH8 was overexpressed, while the W114A mutant did not affect the level of M2 ubiquitination (Fig. 5h, i). We next knocked down (KD) MARCH8 in HEK293T cells and measured the level of M2 ubiquitination. Remarkably, less ubiquitination of M2 was observed in MARCH8 KD cells (Supplementary Fig. 4d, e). Overall, these results demonstrate that MARCH8 ubiquitinates M2, which leads to M2 degradation.

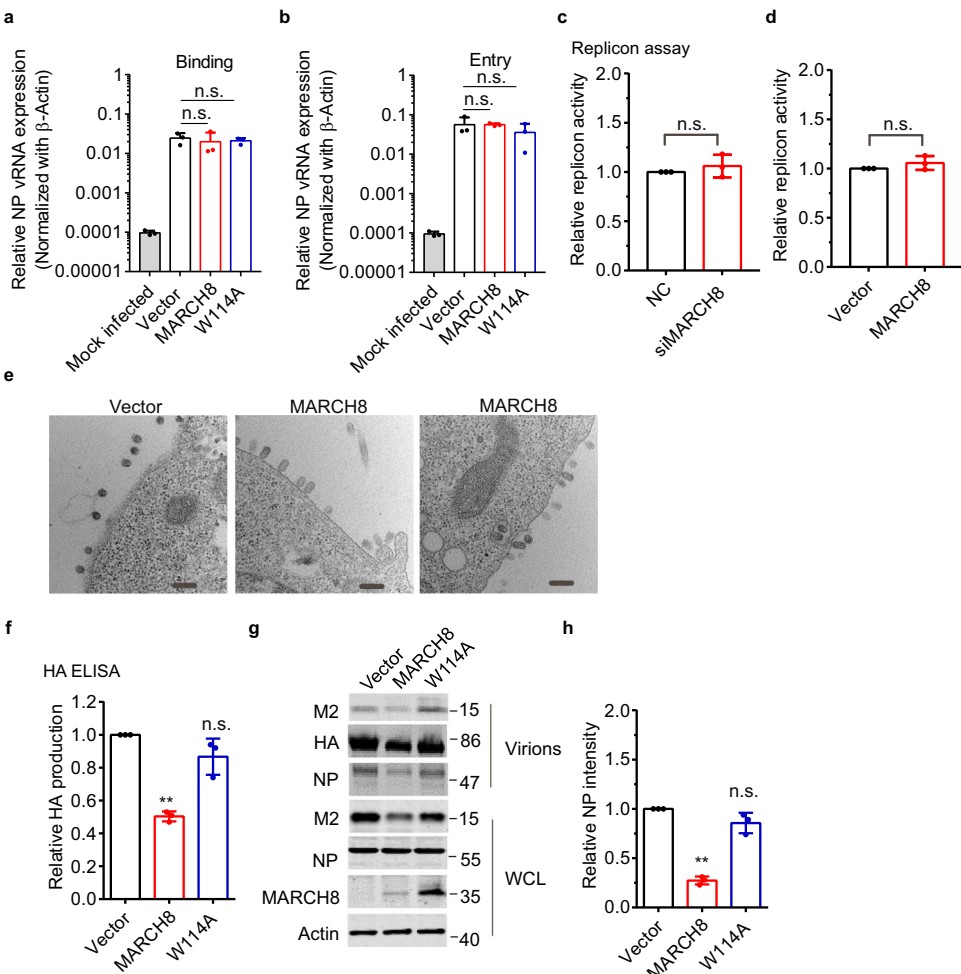

**Fig. 4 MARCH8 reduces IAV release. a**, **b** HEK293T cells were transfected with vector, MARCH8 or W114A. Cells were incubated with WSN virus (MOI = 5) at 4 °C for 1 h (**a**) or then allowed to internalize bound IAV by incubation at 37 °C for another 30 min before adding exogenous NA to remove cell-surface virions. Viral binding or entry was assessed by determining the viral copy number in cell lysates by quantitative real-time PCR. **c**, **d** HEK293T cells were co-transfected with pPoII-Luc and expression plasmids containing viral PB1, PB2, PA, or NP of H1N1 along with control (NC) or MARCH8 siRNAs (**c**), MARCH8 or vector (**d**). Renilla luciferase was used as an internal control. Luciferase activity was determined at 24 h post transfection. **e** HEK293T cells transfected with MARCH8 or vector were infected with an MOI of 5 of WSN virus for 10 h and thin sections were analyzed by electron microscopy. The scale bars indicate 200 nm. **f**–**h** HEK293T cells were transfected with vector, MARCH8 or W114A. Cells were infected with WSN virus at a MOI of 0.2. At 24 h post infection, HA in the supernatants was determined by ELISA (**f**), and viral protein expression in cell lysates and the released virus particles were detected by western blotting (**g**). Quantitation of released NP was shown in (**h**). Data shown are the mean ± SD (n = three independent experiments). **P < 0.001, n.s., nonsignificant, unpaired two-tailed Student t-test, without any adjustments for multiple comparisons.

**MARCH8 mediates K63-linked polyubiquitination of M2 at position K78.** The cytoplasmic domain of M2 contains four conserved lysine residues, K49, K56, K60, and K78, which are potential ubiquitination sites. We therefore substitute each of these four lysines for arginine (Fig. 6a), and examined the expression of these mutated M2 in MARCH8-expressing cells. MARCH8 decreased the expression of the K49R, K56R, K60R mutants to the same extent as that of the WT M2, but did not significantly affect the expression of the K78R mutant and the K/R mutant which has all the above four lysine residues mutated (Fig. 6b, c and Supplementary Fig. 5). Importantly, MARCH8 did not enhance ubiquitination of the K78R and K/R mutants (Fig. 6d). Results of immunofluorescence microscopy further showed that in contrast to the wild-type M2, the cell-surface expression of K78R mutant was not diminished by MARCH8 (Fig. 6e). To determine which type of polyubiquitin linkage to M2 is catalyzed by MARCH8, we transfected M2 and MARCH8 DNA into HEK293T cells together with each of the ubiquitin mutants K6O, K11O, K27O, K29O, K33O, K48O, and K63O, each of

which contains only one lysine available for polyubiquitination. The results of co-immunoprecipitation and western blot showed that only K63-ubiquitin promoted polyubiquitination of M2 as efficiently as the wild-type ubiquitin (Fig. 6f).

**The K78R mutant is more virulent than the wild-type IAV while infecting mice.** To demonstrate how mutation of K78 in M2 affects IAV replication, we generated recombinant WSN and PR8 that bear the K78R M2 mutant and infected the control or MARCH8-expressing A549 cells with the wild-type or the K78R viruses. Indeed, the K78R viruses showed complete resistance to MARCH8 inhibition (Supplementary Fig. 6a). And levels of the K78R M2 protein were not decreased by MARCH8 (Supplementary Fig. 6b). These results suggest that the K78 amino acid in M2 renders IAV sensitivity to MARCH8 inhibition.

We next examined the effect of the K78R mutation in M2 on the virulence of IAV in the mouse model (Fig. 7a). Different doses of IAV, from 20 to 1600 PFU, were used to challenge mice. The median lethal dose (MLD50) was fourfold lower for K78R virus

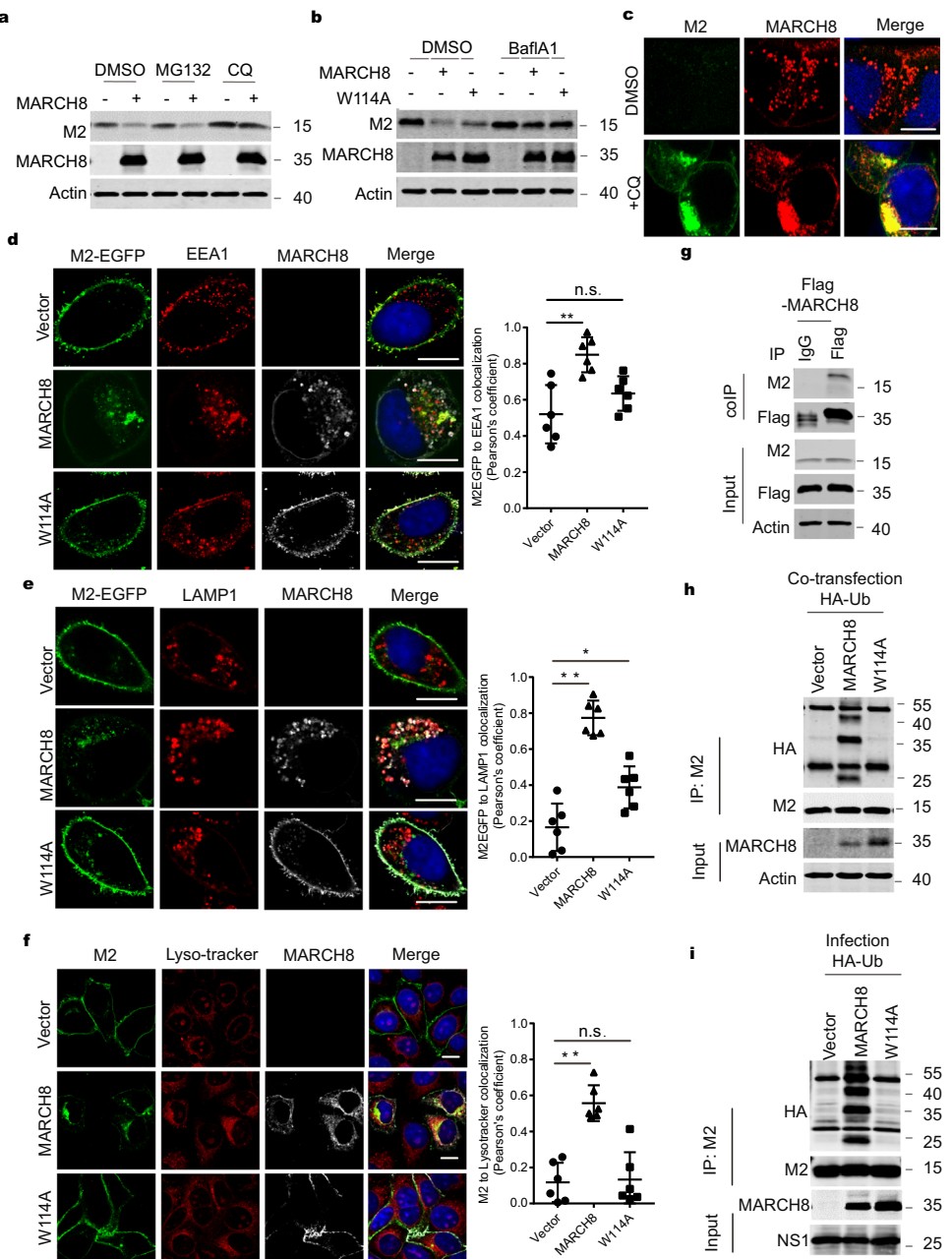

**Fig. 5 MARCH8 ubiquitinates and leads to M2 degradation in lysosomes. a**, **b** Rescue of M2 expression from MARCH8-induced degradation by lysosome inhibitors. HeLa cells were co-transfected with the M2 plus Vector (−), MARCH8 (+), or W114A (+). Cells were treated with DMSO, lysosomal inhibitor chloroquine (CQ, 50 μm) or proteasome inhibitor MG132 (25 μm) (**a**), DMSO or a vacuolar H⁺ translocating ATPase inhibitor (Balf A1, 100 nM) (**b**), followed by western blot. **c** HeLa cells co-transfected with M2 and MARCH8 plasmid were treated with DMSO or CQ and processed for immunofluorescence staining with anti-M2 (green) and anti-MARCH8 antibodies (red). Scale bars represent 10 μm. **d**, **e** HeLa cells transfected with EGFP fused M2 and vector, MARCH8 or W114A were permeabilized, then co-stained with MARCH8 (white) and EEA1 (red) antibodies (**d**) or with MARCH8 (white) and LAMP1 (red) antibodies (**e**). Scale bars represent 10 μm. **f** HeLa cells were co-transfected with M2 plus Vector, MARCH8 or W114A. The lyso-Tracker (red) was added to cells 30 min before fixation. M2 (green) and MARCH8 (white) were detected by immunostaining. Scale bars represent 10 μm. Pearson's correlation coefficient analysis was based on multiple sight fields each group ($n = 6$ fields). *$P < 0.01$; **$P < 0.001$; n.s., nonsignificant, unpaired two-tailed Student $t$-test. **g** Co-immunoprecipitation of M2 and Flag-MARCH8 in HEK293T cells. **h**, **i** MARCH8 ubiquitinates IAV M2. HEK293T cells were transfected with HA-Ub, M2 and Vector, MARCH8 or W114A (**h**). HEK293T cells transfected with HA-Ub and Vector, MARCH8 or W114A were infected with WSN virus (MOI = 2) for 12 h (**i**). Whole-cell lysates were subjected to IP with anti-M2 antibody, and the IP and input were analyzed by western blotting with antibodies against the indicated targets.

(47.7 PFU) than for WT virus (204.1 PFU) (Supplementary Table 1), indicating a marked increase of virulence due to the K78R mutation in M2. At the dose of 200 PFU, K78R IAV-infected mice lost weight more quickly than the wild-type IAV (Fig. 7b), and all of the K78R IAV-infected mice had to be euthanized by day 8, whereas only 40% of the WT virus-infected mice succumbed (Fig. 7c). Associated

with the high pathogenicity, much higher titers of K78R IAV were detected in the lungs of the infected mice compared to those of the wild-type IAV (Fig. 7d). Histopathological assessment of lungs from the infected mice showed that K78R IAV caused more serious bronchiolitis and bronchitis and lumen debris accumulation by 3 days post infection. By day 6, mice infected with K78R virus all

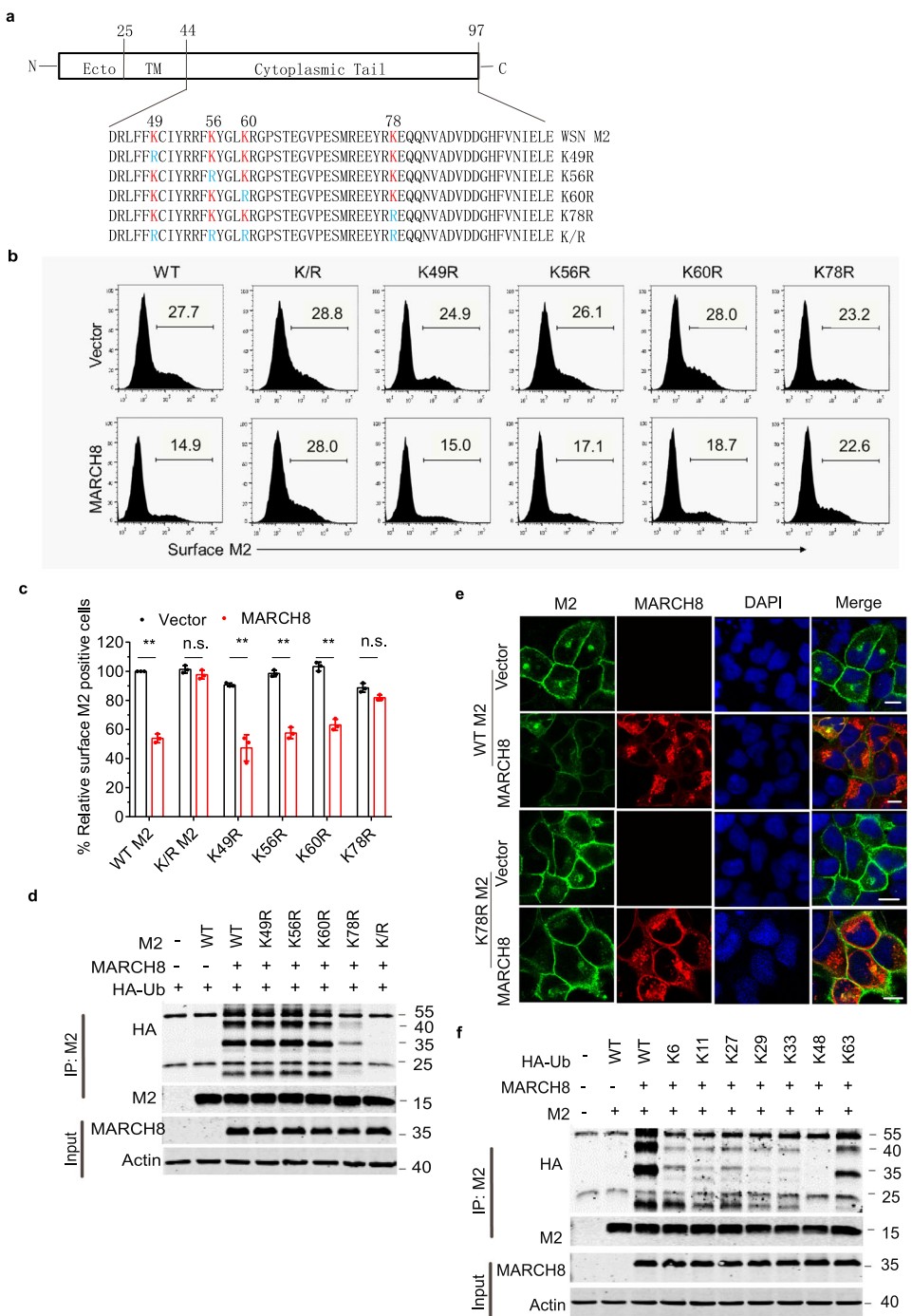

**Fig. 6 MARCH8 mediates K63-linked polyubiquitination of M2 at position K78. a** Illustration of IAV M2 domain-organization and the cytoplasmic tail amino-acid sequence of WSN virus M2 and the indicated mutants. Lysine residues (K) are marked as red, and the K > R mutants are marked as blue. **b, c** HEK293T cells were transfected with M2 mutants and vector or MARCH8. Surface M2 was determined by flow cytometry (**b**). Quantitation of M2 positive cells was shown in (**c**). Data shown are the means ± SD ($n = 3$ biological replicates). **$P < 0.001$; n.s., nonsignificant, unpaired two-tailed Student $t$-test, without any adjustments for multiple comparisons. **d** Ubiquitination of M2 mutants by MARCH8. HEK293T cells were transfected with M2 mutants and vector or MARCH8. Cell lysates were subject to IP with anti-M2 antibody, and the IP and input were analyzed by western blotting with antibodies against the indicated targets. **e** HeLa cells were co-transfected with WT or the K78R mutant M2 and vector or MARCH8. M2 and MARCH8 were detected with specific antibodies in confocal microscopy. Scale bars represent 10 μm. **f** HEK293T cells were transfected with HA-tagged ubiquitin mutants, M2 and MARCH8. Cell lysates were subject to IP with anti-M2 antibody, and the IP and input were analyzed by western blot with antibodies against the indicated targets.

progressed to bronchopneumonia, as opposed to mice infected with WT virus which showed moderate lung inflammation, and only 3 out of 5 mice displayed evidence of bronchopneumonia (Fig. 7e). These results suggest that K78 in M2 is a determinant of IAV virulence.

**Strain-specific degradation of M2 by MARCH8.** When we examined the sequences of M2 cytoplasmic domain from four IAV strains, laboratory-adapted IAV strains WSN and PR8, a 2009 pandemic H1N1 IAV strain (pdm09), and a seasonal H3N2 IAV strain (Supplementary Fig. 7a), we noticed that residue K78

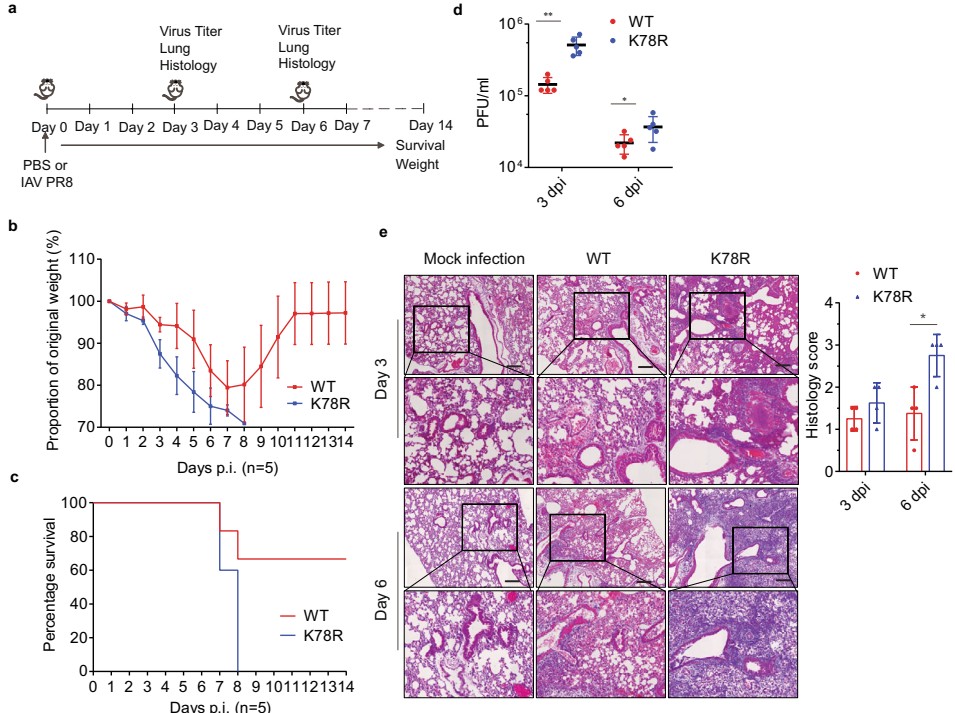

**Fig. 7 Replication of WT and the K78R M2 IAVs in mice.** Six-week-old male C57Bl/6 mice (15 per group) were infected with PR8 virus (WT) or recombinant virus having the K78R M2 mutant (K78R) (200 PFU) intranasally. Five mice per group were euthanized on days 3 and 6 post infection. Bronchoalveolar lavage fluids were collected to measure IAV titers. Lungs were harvested for histopathology. Survival of mice (five per group) was monitored until day 14. **a** Experimental plan. **b** Mouse body weight curves ± SD ($n = 5$ mice) up to day 14 post infection. **c** Mouse survival curves for 14 days ($n = 5$ mice). **d** Mean lung virus titer ± SD ($n = 5$ mice) on day 3 and 6 post infection. **e** Mouse lungs were isolated on day 3 and day 6 post infection. H&E staining was performed on lung sections. Scale bar represents 200 μm. Histological scoring is shown as mean ± SD ($n = 4$ mice). *$P < 0.01$; **$P < 0.001$; n.s., nonsignificant; unpaired two-tailed Student's $t$-test, without any adjustments for multiple comparisons.

is conserved in M2 proteins of WSN, PR8, and H3N2, but not in the pdm09 in which K78 has been changed to Glutamine. Since the M2 protein of pdm09 cannot be detected with M2 monoclonal antibody (C7), we attached the flag tag to the C-terminus of M2 to facilitate detection by western blot (Fig. 8a). When the M2 DNA from each of the above IAV strains was transfected into HEK293T cells together with the MARCH8 DNA, the M2 proteins of WSN, PR8, and H3N2 were ubiquitinated and degraded by MARCH8, while M2 of the pdm09 was resistant to MARCH8 (Fig. 8b, c). We next examined the effect of MARCH8 on the replication of different IAV strains. Replication of the seasonal H3N2 and the two laboratory-adapted IAV strains was markedly inhibited by MARCH8, while the pdm09 strain was not affected (Fig. 8d). These data suggest that different IAV strains have different sensitivity to MARCH8 inhibition.

Since the 1918 H1N1 flu pandemic, the causative H1N1 IAV drifted and reassorted with influenza viruses of other species over time, and has caused several flu pandemics and epidemics in history. We aligned M2 sequences of the IAV epidemics from 1918 until the most recent 2019. Clearly, from 1918 to 1957, human H1N1 IAV strains tended to have K78 in M2, although IAV strains containing non-lysine residues at position 78 slowly emerged over time. This evolving trend became clear in 1979 and 1980 when position 78 in M2 was completely occupied by non-lysine amino acids. While MARCH8 target sites (K78) evolved in H1N1 viruses, this site was persisted in H2N2 and H3N2 viruses (Fig. 8e). Having evolved independently of the former H1N1 virus, the M2 gene of pdm09 H1N1 virus was originated from a Eurasian avian-like swine virus. Historically, swine H1N1 IAV strains have a glutamine at position 78 and a lysine at position 79 in the M2 protein. Starting from 1979, K79 began to change to

E79, which dominates the circulating swine H1N1 since 2010. The pdm09 H1N1 M2 also has Q78 and E79 (Supplementary Fig. 7b). Mutagenesis analysis of 78Q to 78 K or 79E to 79 K showed that K79 functions as the receptor site of ubiquitination in the M2 proteins of pdm09 H1N1 virus (Supplementary Fig. 7c). Overall, these data suggest that in the course of circulating in humans, the H1N1 IAV has gradually changed the K78/K79 sites in the M2 protein to non-lysine amino acids, at least partially to evade MARCH8 restriction.

## Discussion

In contrast to many other enveloped viruses, the viral membrane scission and virion budding of IAV are independent of cellular ESCRT, rather are mediated by viral protein M2[34,35]. Both in vivo and in vitro studies suggest that the amphipathic helix region (residues 45–62) of M2 plays pivotal roles in this function[36,37]. A member of the $Ca^{2+}$ dependent membrane-binding proteins, annexin A6, has been shown to interact with the M2 C-terminal region and interfere with the M2-mediated IAV budding, indicating the presence of cellular mechanisms antagonizing M2 protein[38]. Our study further shows that IAV M2 protein is a substrate of the MARCH8 E3 ligase, and the polyubiquitinated M2 is targeted to lysosome for degradation. Through this mechanism, MARCH8 profoundly inhibits IAV infection both in cultured cells and in mice.

MARCH8 has been reported to inhibit VSV, HIV-1, and other retroviruses by ubiquitinating viral envelope proteins and targeting these key viral proteins for degradation[16]. Interestingly, we did not observe any marked effect of MARCH8 on the levels of IAV HA and NA proteins in both co-transfection and viral infection

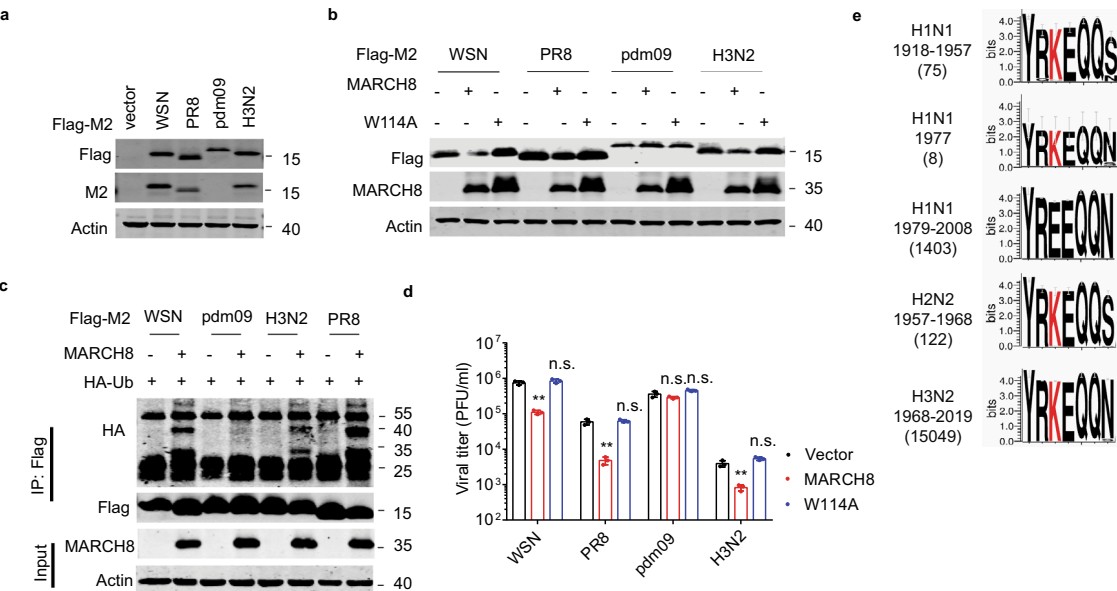

**Fig. 8 Evolution of H1N1 virus to escape MARCH8 restriction. a** HEK293T cells were transfected with Flag-tagged-M2 from different IAV strains. Expression of M2 was examined in western blots. **b** Effect of MARCH8 on the expression of M2 proteins from different IAV strains. The HEK293T cells were transfected with different M2 molecules and Vector, MARCH8 or W114A. M2 and MARCH8 were detected with specific antibodies. **c** Ubiquitination of M2 from different IAV strains by MARCH8. The HEK293T cells were transfected with M2, MARCH8 and HA-Ub. Cell lysates were subject to IP with anti-flag antibody, and the IP and input were analyzed by western blotting with antibodies against the indicated targets. **d** HEK293T cells were transfected with control, MARCH8 or W114A, then infected with WSN virus, PR8 virus, pdm09 virus or H3N2 virus for 24 h (MOI = 0.1). Viral titers in cell culture supernatants were determined by plaque assay. Data shown are the means ± SD ($n = 3$ biological replicates). **$P < 0.001$; n.s., nonsignificant; unpaired two-tailed Student's $t$-test, without any adjustments for multiple comparisons. **e** M2 protein amino-acid sequence logo of IAV isolates. The K78 residues are marked as red. M2 sequences were downloaded from GenBank using the "collapse identical sequences" option. Logos were generated using the WebLogo3 online tool.

experiments. Rather, MARCH8 decreased the expression of viral M2 protein. HA, NA, and M2 are all membrane-bound viral proteins, incorporated into IAV particles, perform different yet essential functions for IAV infection. While HA and NA proteins associate with lipid raft microdomains and initiate viral budding, M2 protein localizes at the neck of the incompletely formed virus particle and mediates membrane scission[28,36]. During IAV infection, M2 has been shown to localize in Golgi during early infection, then be incorporated into the apical vesicles and travels to the plasma membrane[31,39]. Our results suggest that in addition to viral envelope glycoproteins, MARCH8 is also able to target other viral membrane-bound proteins for ubiquitination and degradation, and as a result, restrict viral infection.

M2 protein plays several key roles in IAV replication. First, it is located at the assembly site of IAV and facilitates the scission of viral membrane and subsequent release of the newly formed IAV particles. Indeed, we observed drastic reduction of IAV production in MARCH8-expressing cells, and a marked increase in IAV production when MARCH8 was depleted with siRNA or CRISPR/Cas9. Second, M2 protein can form an ion channel in viral membrane and primes the infectious entry of viral RNA/nucleocapsid complex into the cytoplasm. In addition, when present in cellular membranes, M2 interferes with the pH levels in early endosomes and trans-Golgi network of the secretory pathways[40], thus affecting apical membrane protein trafficking[33], which is believed to be one of the mechanisms leading to rhinorrhea and lung edema, causing exacerbation of respiratory pathology[41]. By restricting the levels of M2 protein in IAV producing cells, MARCH8 may help to mitigate and prevent the cellular pathology associated with M2. Therefore, MARCH8 protects mice from IAV infection not only through direct

reduction of IAV replication but also through mitigation of M2-casued damage to cells and tissues.

Each E3 ubiquitin ligase has its specific group of substrate proteins, which often underlies the function of the E3 ligase in cells including antiviral defense. Our study revealed IAV protein M2 as a viral protein substrate of MARCH8, thus rendering MARCH8 the role of restricting IAV infection and pathogenesis. We also determined the K78, not the other lysine residues in M2, as the main acceptor of polyubiquitin catalyzed by MARCH8. This finding allowed us to examine how IAV evolves over time to evade MARCH8 restriction through changing K78 to other amino acids. Indeed, while K78 in M2 dominated the IAV strains that were isolated in the flu pandemics from 1918 to 1957, this 78 position in M2 was slowly changed to glutamic acid (E78) in M2 prevailed the H1N1 IAV strains since 1978. The H1N1 virus disappeared in the human population in 1957, was replaced by the H2N2 virus, but returned at 20 years later, in 1977[42,43]. Our study reveals one of the mutations, the K78E mutation, that underlies the adaption of the H1N1 IAV to humans and causes its reemergence in 1977.

In our study, the M2-K78R mutation in the PR8 background promotes viral replication in cultured cells and in mice. We further showed that the M2-K78R mutant virus increases M2 accumulation at the cell surface, which was believed to enhance IAV replication based on the previous study reporting that M2 interacts with autophagy factor LC3 at cell surface and augments IAV budding and release[44,45]. We noted that our results are not in agreement with the deleterious effect of M2-K78R mutation on IAV (WSN strain) replication reported by another group[46]. How M2-K78R mutation effect on IAV multiple cycle replication still needs further study.

A number of innate restriction mechanisms have been discovered that protect mice from lethal IAV infection. One of the first findings in this regard is the interferon-induced myxovirus resistance (Mx) genes that are often defective in mice which are commonly used in laboratories and succumb to IAV infection[47]. In addition to Mx proteins that target IAV nucleocapsid and inhibit viral RNA transcription[48–50], IFITM proteins provide another layer of defense against IAV infection by deterring viral entry[49,51,52]. *MARCH* genes are not induced by interferon, though *Mx* and *IFITM* genes are interferon-stimulated genes. Results of our study demonstrated that deletion of MARCH8 renders mice to succumb to otherwise non-lethal dose of IAV infection, thus uncovering a distinct key intrinsic immune mechanism protecting mice from lethal IAV infection.

In summary, our data support the potent restriction of IAV infection by MARCH8 both in tissue culture and in the mouse model. In contrast to other known anti-IAV innate immune factors, MARCH8 acts by targeting IAV M2 protein for ubiquitination-mediated degradation in lysosomes, thus profoundly inhibiting IAV release and production. The inhibitory pressure of MARCH8 on IAV is further supported by evasion of IAV from MARCH8 over the prolonged period of circulation in humans.

## Methods

**Plasmids and reagents.** Human MARCH8-expression plasmids pQCXIP-MARCH8 and its RING-CH domain mutant pQCXIP-MARCH8-W114A were constructed by inserting the *MARCH8* cDNA into the *Not* I/*Bam*H I sites in the pQCXIP retroviral vector (Clontech). The pcDNA3.1 vectors encoding HA, NA and M2 of A/WSN/33(H1N1) were generated as previously described[53]. M2 DNA sequences from laboratory-adapted influenza virus A/WSN/33(H1N1) (CAA30883.1) and A/PR/8/1934(H1N1) (NP_040979.2), pandemic influenza virus A/Hamburg/4/2009(H1N1) (ACR10231.2) and seasonal influenza virus A/Panama/2007/1999(H3N2) (ABE73112.1) were synthesized and subcloned into the pCMV-Tag 2B vector.

Antibodies used for western blot are as follows: MARCH8 (Proteintech; Cat# 14119-1-AP; 1:1000), M2 (Santa Cruz; Cat# SC-32238; 1:2000), H1N1 HA (Sino Bio.; Cat# 11684-RP01; 1:2000), P24 (Sino Bio.; Cat# 11651-RP01; 1:5000), VSV-G (Sigma-Aldrich; Cat# V5507; 1:2000), Flag (Sigma-Aldrich; Cat# F7425; 1:2000), Myc (Sigma-Aldrich; Cat# C3956; 1:2000), NP (Millipore; Cat# MAB8251; 1:1000), NA, NS1, PA, PB1, PB2 (GeneTex, Cat#629696, 1:2000; Cat# GTX125990, 1:2000; Cat# GTX125932, 1:2000; Cat# GTX125923, 1:2000; Cat# GTX125926, 1:2000; respectively), HA.11 Epitope Tag (BioLegend, Cat# 901501 or Cat# 902301; 1:1000), and actin (Sigma-Aldrich, Cat# A1978).

**Cells, viruses, and animals.** Human embryonic kidney cells (HEK293T), human lung adenocarcinoma cells (A549), human cervical cancer cells (HeLa), and Madin-Darby canine kidney cells (MDCK) were purchased from the American Type Culture Collection (ATCC). All cell lines were maintained in Dulbecco's modified Eagle medium (DMEM) supplemented with fetal calf serum (FCS 10% v/v, HyClone) and penicillin–streptomycin (100 U/ml, Thermo Fisher Scientific).

To generate respiratory epithelial cell lines that stably express MARCH8, A549 cells were transduced with pQCXIP (Clontech)-based pseudovirus according to the manufacturer's instruction. To generate MARCH8-knockout cell lines, a lentiviral CRISPR-Cas9 expression plasmid pCRISPR-MARCH8 was created by inserting DNA fragments that contain a target sequence of MARCH8 (1#: 5′-GTAAGACCAAAGAAAAGGAG-3′ or 2#: GAGCTCGCAGCAGCGCGTGT) into lentiCRISPR-v2 (Addgene). Cells were transduced with pseudo-viruses expressing ctrl CRISPR-Cas9 or CRISPR-Cas9 targeting MARCH8. Stably transduced cell lines were selected with puromycin, and cell clones were generated with limited dilution. Single clonal cell lines were screened via western blot.

The M2-K78R mutant viruses mutPR8 and mutWSN were generated via the reverse genetic system as described previously[54]. The M segment of PR8 or WSN virus with M2-K78R mutant was generated by site-directed mutagenesis in the vRNA-mRNA bidirectional transcription vector pBD. Viruses were generated by transfecting HEK293T and MDCK co-cultured cells with the 8 plasmids. WSN, mutWSN, pdm09 H1N1 (A/Beijing/01/2009) and H3N2 (A/Beijing/30/95) viruses were propagated in MDCK cells in minimum essential medium (MEM) supplemented with 0.5% bovine serum albumin (BSA) in the presence of 1 μg/ml tosylphenylalanyl chloromethyl ketone (TPCK)-treated trypsin, 1% (v/v) penicillin/streptomycin. Supernatants from the virus cultures were harvested 3 days post infection. Virus stocks were aliquoted and stored at −80 °C. Viral titers were calculated using plaque assays. The PR8 and mutPR8 viruses were inoculated into and propagated in 9-day-old specific pathogen-free (SPF) chicken embryos for 3 days and then harvested and stored at −80 °C until use.

Six-week-old C57Bl/6 background male mice were purchased from Beijing Vital River Co. Ltd. (Beijing, China). Animal studies were carried out in specific pathogen-free barrier facilities. Facilities were maintained at an acceptable range of 20–26 °C humidity of 30–70% on a 12-h dark/12-h light cycle.

**IAV binding, entry assay.** HEK293T cells were seeded into 6-well plates at $5 \times 10^5$ cells per well and cultured for 24 h. Cells were transfected with the vector, MARCH8 or W114A expression plasmids. After 24 h, cells were infected with IAV (MOI = 5) and incubated at 4 °C for 1 h. For the virus binding assay, cells were washed with cold PBS (at 4 °C) twice to remove unbound virus and cell lysates were harvested, the amount of viral RNA was determined by RT-PCR. For virus entry assay, after incubation at 4 °C for 1 h to allow for viral binding, infected cells were washed with cold PBS (at 4 °C) twice to remove unbound virus, followed by incubation with pre-warmed DMEM for 30 min at 37 °C. Subsequently, cells were treated with neuraminidase (Merck) for another 30 min at 37 °C and rinsed three times with PBS to remove the attached but not yet internalized virions. Total cellular RNA was extracted and quantified by RT-PCR.

**IAV minigenome system for polymerase activity.** HEK293T cells were seeded into a 6-well plate at $5 \times 10^5$ cells per well. After 24 h, cells were transfected with pcDNA3.1-PB1, -PB2, -PA (100 ng each) and -NP (300 ng), luciferase reporter pPolI-NP-Luc (100 ng), and pTK-RL (10 ng), and vector or MARCH8-expression plasmids or control or MARCH8 siRNAs using Lipofectamine 2000 (Invitrogen). Cells were lysed and analyzed with a dual-luciferase reporter assay kit according to the instructions of the manufacturer (Promega).

**Real-time quantitative PCR (qPCR) analysis.** Total cellular RNA was extracted using the PureLink RNA Extraction kit (Thermo Fisher Scientific) according to the manufacturer's protocol. cDNA was synthesized using the PrimeScript™ RT reagent Kit with gDNA Eraser (TaKaRa). A *NP* vRNA specific primer was used for reverse transcription of vRNA of *NP*. An oligo(dT)20 primer was used for reverse transcription of cellular mRNA. *β-actin* mRNA was used as an internal control. The sequences of primers used in qPCR reactions were provided in Supplementary Table 1. Real-time qPCR assays were further analyzed by Bio-Rad CFX Manager.

**Electron microscopy.** For thin-section electron microscopy, WSN IAV-infected HEK293T cells were scraped off the plates. Cells were then centrifuged at $1000 \times g$ for 10 min to form a pellet. Cell pellets were fixed with 2% paraformaldehyde–2.5% glutaraldehyde solution for at least 4 h and then fixed with 1% osmium tetroxide for 1.5 h. Specimens were subsequently dehydrated in gradient ethanol. The cells were further embedded in epoxy resin PON812 and polymerized at 60 °C for 24 h. Eighty-nanometer-thickness sections were obtained from the resin blocks and were placed on copper grids and stained with uranyl acetate and lead citrate. The negative stained ultrathin sections were observed under the transmission electron microscope.

**VLP production and purification.** VLPs were produced as previously described[53]. Briefly, cells were seeded in 10-cm dishes in complete DMEM for 24 h. Then, cells were co-transfected with indicated viral protein expression plasmids for 48 h. The culture supernatants were collected and passed through a 0.45-μm filter. VLPs were pelleted by ultracentrifugation through a 20% sucrose cushion ($200,000 \times g$ for 3 h at 4 °C, Beckman SW41Ti rotor).

**IAV infection and purification.** For virus infection, cells were first washed twice with PBS (Gibco) before inoculated with diluted virus at indicated MOI for 1 h at RT. After washing twice with PBS, cells were cultured in DMEM-BSA containing 0.5 μg/ml tosylsulfonyl phenylalanyl chloromethyl ketone (TPCK)-treated trypsin. To harvest IAV particles, the culture supernatants were filtered through a 0.45-μm filter and virus particles were pelleted by ultracentrifugation through a 20% sucrose cushion ($200,000 \times g$ for 3 h at 4 °C, Beckman SW41Ti rotor).

**HA protein ELISA.** The concentration of released HA protein was measured in clarified cell culture supernatants by ELISA (Sino Biological) against standard curves of recombinant PR8 virus HA antigen, according to the manufacturer's instructions.

**Plaque assay.** To determine the plaque-forming unit, viral supernatants were collected and plaque titers were determined by the plaque assay. Briefly, monolayer MDCK cells in 12-well plates were incubated with serial dilutions (10 times) of viral supernatants in 0.25 ml at room temperature (RT) for 1 h with swirling every 15 min. One milliliter of 1% agarose with 0.25% fetal bovine serum was then added to the cells and left at RT until it set. Then the dishes were turned upside down and incubated at 37 °C. At 72 h post infection, the agarose layer was removed and the plaques were visualized with 0.1% crystal violet solution.

**Flow cytometry.** Flow cytometry analysis was performed as previously described[55]. In brief, cells were detached and resuspended in complete growth media. Cells were

fixed with 4% paraformaldehyde for 10 min at room temperature. Cells were then incubated for 30 min on ice with an anti-Influenza A M2 antibody (Santa Cruz; Cat# SC-32238; 1:100), followed by staining for 30 min on ice with a rabbit anti-mouse IgG conjugated with Alexa Fluor 488 (Thermo Fisher; A21446; 1:1000). Cells were analyzed for the expression of surface M2 using BD FACSCanto II Flow Cytometer. The data were collected and analyzed with BD FACSDiva™ Software v8.0 software and Flowjo v10.

**Ethics statement**. The animal experiments were performed according to the Chinese Regulations of Laboratory Animals-The Guidelines for the Care of Laboratory Animals (Ministry of Science and Technology of People's Republic of China) and Laboratory Animal-Requirements of Environment and Housing Facilities (GB 14925 ± 2010, National Laboratory Animal Standardization Technical Committee). The license number associated with this research protocol was CAU 20190611, which was approved by the Institute of Animal Use and Care Committee of the Institute of Laboratory Animal Science, Peking Union Medical College.

**Infection of mice**. The mouse-adapted strain of influenza A/Puerto Rico/8/1934 (PR8) was propagated in 10-day embryonated eggs and titered with a plaque assay. C57Bl/6 background male mice aged 6–8 weeks were purchased from Vital River Laboratory Animal Technology (Beijing). Mice were anesthetized by intraperitoneal injection of a mixture of Ketamine and Xylazine (100 and 5 μg per gram of body weight). Mice were infected intranasally with 200 pfu of viruses in 40 μL PBS and monitored daily for weight loss and clinical signs. Mice were euthanized if they had ≥30% loss of initial body weight. To analyze viral release, bronchiolar alveolar lavage fluid (BALF) was collected and processed from infected and uninfected mice by postmortem lavage with $2 \times 300$ μL ice-cold PBS as described[31]. For histopathologic analysis, lungs were removed on day 3 of day 6 post infection and fixed with 4% PFA. Lung samples were embedded in paraffin and cut into 5-μm-thick sections, followed by staining with haematoxylin and eosin (H&E). For the lung histology score, images were evaluated by an investigator in a blinded manner following a standardized score system as previously described[56]. Lung injury scoring system included five different parameters: neutrophils in alveolar airspace, neutrophils in the interstitial space, hyaline membranes, proteinaceous debris filling the airspace, and alveolar septal thickening.

**MARCH8 knockdown by PPMOs**. PPMOs were purchased from Gene Tools. PPMO targeting mouse *MARCH8* gene (PPMO-M8) was designed as CATGCT-CATCCCAGCCTCCGAC. A nontargeting PPMO control sequence (PPMO-NC) (CCTCTTACCTCAGTTACAATTTATA), having little homology to mouse transcripts or influenza viral sequences, was used as control. Six to eight weeks C57Bl/6 male mice were inoculated intranasally with 100 μg of PPMOs in 40 μl PBS for 2 days continuously. Lung homogenates were prepared using a FastPrep24 system (MP Biomedicals). After addition of 800 μl of PBS containing 0.3% BSA, lungs were subjected to two rounds of mechanical treatment for 10 s each at 6.5 m/s. Tissue debris was removed by low-speed centrifugation, and virus titers in supernatants were determined by plaque assay.

**Immunofluorescence microscopy**. IF was performed with HeLa cells 24 h post transfection with IAV M2 and MARCH8 plasmid DNA. For lysosome labeling, lyso-Tracker-Red99 (Invitrogen) was used to treat cells 24 h post transfection of M2 DNA and incubated for 30 min prior to IF staining. Early or late endosomes were labeled by anti-EEA1 (BD Biosciences; Cat# 610456; 1:50) or anti-LAMP-1 (BioLegend; Cat# 328602; 1:250) antibody or via transfection of a plasmid encoding Rab5-GFP or Rab7-GFP. Confocal images were collected with Leica TCS SP5 confocal microscope, and analyzed by LAS AF Lite v3.0 and ImageJ v1.44.

**Detection of ubiquitination by IP**. Cells transfected with IAV M2 or infected with IAV and control cells were treated with CQ (50 μM) for 4 h and lysed in a buffer containing 100 mM Tris-HCl (pH 8.0), 0.15 M NaCl, 5 mM EDTA, 1% NP-40, 0.5% Triton X-100, DUB inhibitors (100 mM PR619, 5 mM 1,10-phenanthroline, 5 mM NEM), and a protease inhibitor cocktail. Lysates were spun at $12,000 \times g$ for 10 min. 300 μl reaction buffer (100 mM Tris-HCl (pH 8.0), 0.15 M NaCl, 5 mM EDTA) was added to 300 μl lysis buffer. Anti-M2 or IgG antibodies were then added to the clarified supernatants for 2 h followed by A/G Dynabeads and 16 h incubation at 4 °C. Beads were washed with catch and release IP wash buffer (Invitrogen), and eluted in 2xSDS sample buffer. The samples were boiled and separated by SDS-PAGE, then wet transferred to a 0.45 μm PVDF membrane (Millipore). The western blot images were collected and analyzed with LICOR Odyssey CLx with ImageStudio lite v5.2 software.

**Selection of M2 sequences for alignment**. The sequences of M2 proteins from human IAVs circulating during different times were selected from the Influenza Research Database (IRD). The strain selection was refined, discarding M2 sequences that were identical and only considering fully sequenced M2. In addition, HA/NA subtype and year were also defined. Following these definitions, lists of the most representative M2 were created. Then, aligned M2 protein sequences were uploaded to the WebLogo3 online tool, and sequence logo was generated.

**Statistical analysis**. All data were plotted and statistical analyses performed with GraphPad Prism 7.0 software (GraphPad). Unless otherwise indicated, graphs display mean ± standard deviation (SD) and represent data from at least three independent experiments. Statistical significance was analyzed using two-tailed unpaired Student's *t*-test. Significance indicated by asterisks is designated as follows: *$p < 0.01$; **$p < 0.001$; n.s., nonsignificant.

**Statistics and reproducibility**. For in vitro experiments, phenotypic analyses including immunofluorescence, western blot, and FASC were performed in at least three independent experiments, using biological replicates.

## Data availability

M2 sequences can download from the Influenza Research Database (https://www.fludb.org/). Source data are provided with this paper.

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

## Acknowledgements

We would like to thank Dr. Zhaohui Qian (IPB, Beijing) for kindly provide MARCH8 cDNA. We would like to thank Jian Rao (IPB, Beijing) for assisting in the in vivo experiment. We are grateful to Li Li (IPB, Beijing) for technical assistance in performing confocal microscopy. Furthermore, we would like to thank Adolf García-Sastre from Icahn School of Medicine at Mount Sinai for helpful discussion. Finally, we would like to thank all members of the Zhaohui Qian's lab and Tao Deng's lab of Institute of Pathogen Biology for useful discussion. This study was supported by funds from the National Key Plan for Scientific Research and Development of China (2018YFE0107600, 2016YFD0500307, and 2020YFA0707600), from the Ministry of Science and Technology of China (2018ZX10301408-003 and 2018ZX10731101-001-018), from the National Natural Science Foundation of China (82072288, 81371808, 81528012, and 81401673), from CAMS Innovation Fund for Medical Sciences (CIFMS 2018-I2M-3-004, 2020-I2M-2-014, CIFMS 2016-I2M-1-014), from the Canadian Institutes of Health Research (CCI-132561), and from the CAMS general fund (2019-RC-HL-012).

## Author contributions

Z.Z., S.C., C.L., J.W. and F.G. conceived the project. X.L., F.X., L.R., F.Z., Y.H., L.W., Z.F., S.M. and J.S. performed the experiments. C.W. and Y.W. performed the in vivo experiment. All authors contributed to experimental design and data analysis. X.L., C.L. and F.G. composed the manuscript. All authors reviewed the manuscript and discussed the work.

## Competing interests

The authors declare no competing interests.
