## [Peer Review File · Nature Communications]

Reviewer comments, first round –

Reviewer #3 (Remarks to the Author):

Minor comment:

Figure 8e, main text lines 298-316:

It is very interesting that the K78E mutation emerged in 1978/79. The H1N1 virus was substituted by H2N2 virus in the human population in 1957 and reappeared in 1977 after a 20-year absence, first in China and Russia (Russian flu). The origin of reintroduction of the virus into the human population is still unclear. The virus was genetically very similar to an isolate from 1950, suggesting that it could have evolved slowly in non-human hosts since 1957 or was kept frozen in a yet unidentified source. The authors should implement the 1977 re-appearance of the H1N1 virus in the discussion.

Reviewer #4 (Remarks to the Author):

In this manuscript Liu, Xu, Ren and colleagues convincingly demonstrate that M2 is a target of MARCH8 ubiquitination that results in the degradation of M2 and subsequent anti-viral effects both in vitro and in vivo. The finding that MARCH8 inhibits influenza replication is novel, and the fact that it acts by degrading the viral M2 protein rather than HA or NA is also intriguing given its mode of action on other viruses. This study maps the ubiquitination of MARCH8 to lysine residue 78, and shows specific deficits at the stage of viral budding/particle release. The work is clearly presented and should be complemented for its thoroughness and use of complementary orthogonal approaches. Clarifying the following issues will strengthen the stated conclusions and some areas where the data is difficult to interpret.

- It is a little hard to reconcile the near complete loss of M2 production seen by western blot (Fig 1e) with the fact that there was only a two-fold drop in surface M2 (Fig1G) and similar levels of M2 intensity are seen in the Vector and MARCH8 panels of Fig 1H. Was the microscopy all collected at equal laser/PMT settings? I am convinced that MARCH8 is degrading M2, but less sure about the specific effect on cell surface localization (the distribution of M2 does not appear to be particularly localized to the cell surface at 10 hours even in cells with only vector, for instance).
- Scoring of histology from multiple animals rather than a single representative image would make this data more robust (Fig 3d and 7e).
- It is difficult to interpret the viral entry data from fig 4a, as an acid wash step will force fusion at the plasma membrane and result in the incorporation of viral RNA into the cell from virions that were unable to endocytosis. If the authors use a protease (such as exogenous NA) to remove background binding this would give them a more accurate picture of the levels of viral endocytosis. It would also be helpful to report the levels of viral binding (by harvesting samples after the one hour incubation at 4 degrees before warming). Related- the conclusion sentence (lines 177-179) should be modified- even after swapping the acid wash for protease treatment these two assays will only measure binding, entry, viral transcription, replication and translation. The entry assay measures viral RNA regardless of whether uncoating has happened, and as the mini-replicon assay uses transfected plasmids it is not suitable to measure vRNP nuclear import.
- In Fig 5 the authors conclude that MARCH8 is redistributing M2 from the plasma membrane to endosomes and lysosomes but the quality of the EEA1 and Lyso-tracker staining makes this hard to see- both EEA1 and lyso-tracker appear as diffuse, faint staining throughout the cytoplasm. The internal M2-EGFP staining in cells expressing MARCH8 in Fig 5e,d,f doesn't seem to colocalize with MARCH8 which is hard to square with the image in Fig 5c- is there something about adding GFP to M2 that means it no longer interacts with MARCH8? In Supp Fig 4 the Rab7/MARCH8 colocalization is much more convincing than the Rab5/MARCH8 colocalisation. For all this imaging it would help to generate pearson colocalization coefficients from many fields of view to increase the robustness

of the observation.

- The authors should discuss how their data fits with that published by Su, Yu, Huang, & Lai (JVI, 2017)- specifically the fact that the K78 mutation in the WSN background reduced rather than increased titre- but also how ubiquitination at residue 78 controls autophagy. Related- were the viruses created in this study sequenced to confirm the presence of the desired mutation (and absence of any secondary mutations?)

Minor comments

- Fig 1 d, the top +/- section should be labelled MARCH 8 WT or MARCH8 W114A for clarity.
- Fig2b,e,h,c,f,I should have the same y axis units.
- The electron microscopy presented in Fig 4e does not show the classical 'beads on a string' phenotype, as that would involve multiple virions in a chain originating from a single budding site. What is shown here does appear to be a large number of virions that are nearly finished budding but remain attached/adjacent to the plasma membrane (ie, a budding defect)- but it is not strictly speaking "beads on a string".
- What about NP trafficking? Disrupting NP trafficking would also affect viral budding.
- Line 202: decrease should be decreases.
- It is difficult to see the M2 signal in Supp 4b,c, d when MARCH8 is expressed, which makes interpreting the suggested colocalization difficult . It would also be helpful to quantify this phenotype in more cells and provide a higher magnification/resolution image of the colocalization.
- Fig 5G, labelling MARCH8 as Flag, or Flag-M8 is confusing (I'm assuming a flag tagged MARCH8 is used as bait here?)
- There appears to be a 14 kDa ubiquitin modification on M2 that is not MARCH8 dependent (see band at 40 kDa in vector and M114A conditions in Fig 5i)- can the authors comment on how this may affect their conclusions?
- Line 243, ubiquitination is spelled wrong.
- In the discussion (line 319), the correct references for influenza virus ESCRT independence are Bruce et al (pmid 19524996) and Chen et al (pmid 17475660).
- The information on generation of mutant viruses is in two different sections in the methods (Cells, viruses and animals and Generation of mutant viruses) and this should be combined into a single section for clarity.
- The methods for the entry assays says attached but not internalized virions were removed by washing with PBS, the figure legend says they were removed with acid wash- which method was used?
- Please provide references for the reverse genetics system used.
- What are the primer-probe sequences used for RT-PCR detection?
- For electron microscopy were the cells fixed prior to scraping?
- If electron tomography rather than standard transmission electron microscopy was performed can the authors provide the 3D renderings/movies? Alternatively clarify in the methods which type of microscopy was performed.

Reviewer #5 (Remarks to the Author):

The reviewer thanks Guo and colleagues for satisfactorily revising the manuscript to address concerns raised by this reviewer. The revised manuscript requires careful proofreading before publication.

REVIEWER COMMENTS

Reviewer #3 (Remarks to the Author):

Minor comment:

Figure 8e, main text lines 298-316:

It is very interesting that the K78E mutation emerged in 1978/79. The H1N1 virus was substituted by H2N2 virus in the human population in 1957 and reappeared in 1977 after a 20-year absence, first in China and Russia (Russian flu). The origin of reintroduction of the virus into the human population is still unclear. The virus was genetically very similar to an isolate from 1950, suggesting that it could have evolved slowly in non-human hosts since 1957 or was kept frozen in a yet unidentified source. The authors should implement the 1977 re-appearance of the H1N1 virus in the discussion.

Response: As suggested by this reviewer, we have now discussed the 1977 re-appearance of the H1N1 virus in lines 372 to 376, “The H1N1 virus disappeared in the human population in 1957, was replaced by the H2N2 virus, but returned at 20 years in 1977. Our study reveals one of the mutations, the K78E mutation, that underlies the adaption of the H1N1 IAV to humans and causes its reemergence in 1977.”

Reviewer #4 (Remarks to the Author):

In this manuscript Liu, Xu, Ren and colleagues convincingly demonstrate that M2 is a target of MARCH8 ubiquitination that results in the degradation of M2 and subsequent anti-viral effects both in vitro and in vivo. The finding that MARCH8 inhibits influenza replication is novel, and the fact that it acts by degrading the viral M2 protein rather than HA or NA is also intriguing given its mode of action on other viruses. This study maps the ubiquitination of MARCH8 to lysine residue 78, and shows specific deficits at the stage of viral budding/particle release. The work is clearly presented and should be complemented for its thoroughness and use of complementary orthogonal approaches. Clarifying the following issues will strengthen the stated conclusions and some areas where the data is difficult to interpret.

• It is a little hard to reconcile the near complete loss of M2

production seen by western blot (Fig 1e) with the fact that there was only a two-fold drop in surface M2 (Fig1G) and similar levels of M2 intensity are seen in the Vector and MARCH8 panels of Fig 1H. Was the microscopy all collected at equal laser/PMT settings? I am convinced that MARCH8 is degrading M2, but less sure about the specific effect on cell surface localization (the distribution of M2 does not appear to be particularly localized to the cell surface at 10 hours even in cells with only vector, for instance).

Response: We also noticed that the level of M2 decreased more dramatically in the western blot compared to the decrease of cell surface M2. One possible explanation is the different detection sensitivity of western blot and immunofluorescence staining. It is also possible that some of the membrane-bound M2, especially with MARCH8 expression is resistant to protein extraction, thus lost and not detected in western blot.

We have now provided higher resolution images of M2 localization in Fig. 1h. Immunostaining of M2 at 6 hrs, 8 hrs and 10 hrs post-infection showed that M2 was mainly localized to the cell surface after synthesis (Fig. 1h, cells transfected with only vector), which is consistent with the results by other groups showing that the newly synthesized M2 protein trafficks from the ER to the Golgi complex and then to the plasma membrane. Since MARCH8-overexpressing cells exhibited undetectable M2 at the cell surface, but comparable intracellular M2 protein level with control cells (Fig. 1h, cells transfected with MARCH8), we conclude that MARCH8 prevents the localization of M2 to the cell periphery. The microscopy images were all collected at equal laser/PMT settings.

• Scoring of histology from multiple animals rather than a single representative image would make this data more robust (Fig 3d and 7e).

Response: We have now provided all histology images of mouse lung tissue sections collected for the studies shown in Fig 3d and 7e. Lung tissue sections from each mice were numbered. The images are scored and the data are included in Fig 3d and 7e.

• It is difficult to interpret the viral entry data from fig 4a, as an acid wash step will force fusion at the plasma membrane and result in the incorporation of viral RNA into the cell from virions that were unable to endocytosis. If the authors use a protease (such as exogenous NA) to remove background binding this would give them a more accurate picture of the levels of viral endocytosis. It would also be helpful to report the levels of viral binding (by harvesting samples after the one hour incubation at 4 degrees before warming). Related- the conclusion sentence (lines 177-179) should be modified- even after swapping the acid wash for protease treatment these two assays will only measure binding, entry, viral transcription, replication and translation. The entry assay measures viral RNA regardless of whether uncoating has happened, and as the mini-replicon assay uses transfected plasmids it is not suitable to measure vRNP nuclear import.

Response: We have measured the level of virus binding as suggested by this reviewer, the data are presented in Fig. 4A. We also repeated the virus entry assay using exogenous NA, the data are presented in Fig. 4B. Methods of these two assays were described in lines 460 to 472, “**IAV binding, entry assay.** HEK293T cells were seeded into 6-well plates at 5×10^5 cells per well and cultured for 24 hours. Cells were transfected with the vector, MARCH8 or W114A expression plasmids. After 24 hours, cells were infected with IAV (MOI = 5) and incubated at 4°C for 1 hour. For the virus binding assay, cells were washed with cold PBS (at 4°C) twice to remove unbound virus and cell lysates were harvested, the amount of viral RNA was determined by RT-PCR. For virus entry assay, after incubation at 4°C for 1 hour to allow for viral binding, infected cells were washed with cold PBS (at 4°C) twice to remove unbound virus, followed by incubation with pre-warmed DMEM for 30 min at 37°C. Subsequently, cells were treated with neuraminidase (Merck) for another 30 min at 37°C and rinsed three times with PBS to remove the attached but not yet internalized virions. Total cellular RNA was extracted and quantified by RT-PCR.” We now conclude in lines 177 to 179, “Taken together, these data suggest that MARCH8 does not affect the early stages of the viral replication including virus binding, virus entry, and viral RNA transcription, replication and translation.”

• In Fig 5 the authors conclude that MARCH8 is redistributing M2 from the plasma membrane to endosomes and lysosomes but the quality of the EEA1 and Lyso-tracker staining makes this hard to see- both EEA1 and lyso-tracker appear as diffuse, faint staining throughout the cytoplasm. The internal M2-EGFP staining in cells expressing MARCH8 in Fig 5e,d,f doesn't seem to colocalize with MARCH8 which is hard to square with the image in Fig 5c- is there something about adding GFP to M2 that means it no longer interacts with MARCH8? In Supp Fig 4 the Rab7/MARCH8 colocalization is much more convincing than the Rab5/MARCH8 colocalisation. For all this imaging it would help to generate pearson colocalization coefficients from many fields of view to increase the robustness of the observation.

Response: We have repeated the staining of EEA1 and LAMP1, and presented the new images in fig 5d and fig 5e. We have also calculated the co-localization Pearson's coefficients, the data are now presented in Fig. 5d, 5e, 5f and in Sup Fig 4c, 4d.

In Fig 5c, an M2 monoclonal antibody was used to immunostain the M2 protein, while in fig. 5d and 5e M2-EGFP fusion protein was used to visualize M2 localization. The weaker co-localization of M2-EGFP with MARCH8 (fig 5e and 5d) might be the result of low fluorescence of EGFP at low pH, i.e. in lysosomes where M2-EGFP is recruited by MARCH8 (fig 5c).

• The authors should discuss how their data fits with that published by Su, Yu, Huang, & Lai (JVI, 2017)- specifically the fact that the K78 mutation in the WSN background reduced rather than increased titre- but also how ubiquitination at residue 78 controls autophagy. Related- were the viruses created in this study sequenced to confirm the presence of the desired mutation (and absence of any secondary mutations?)

Response: Su, Yu, Huang, & Lai (JVI, 2017) constructed an M2-K78R mutant virus in the WSN background, and observed that this mutated virus contained more HA proteins, less viral RNAs, M1 and NP, than the wild-type virus, and was thus impaired in infection and replication. In our study, we observed that the M2-K78R mutant virus in the PR8 or WSN background had similar levels of HA, M1 and NP proteins to those in the wild type virus, and produced more infectious progeny particles. One possible explanation of this discrepancy is that the M2-K78R mutation was engineered with different IAV strains in these two studies. We did sequence the virus we generated and confirmed that there were no secondary mutations introduced. We have now discussed this issue in lines 378 to 385, "In our study, the M2-K78R mutation in the PR8 background promotes viral replication in cultured cells and in mice. We further showed that the M2-K78R mutant virus increases M2 accumulation at the cell surface, which is believed to enhance IAV replication based on the previous study reporting that M2 interacts with autophagy factor LC3 at cell surface and augments IAV budding and release. We noted that our results are not in agreement with the deleterious effect of M2-K78R mutation on IAV (WSN strain) replication reported by another group. How M2-K78R mutation effect on IAV multiple cycle replication still need further study."

Minor comments

• Fig 1 d, the top +/- section should be labelled MARCH 8 WT or MARCH8 W114A for clarity.

Response: This error has now been corrected in fig.1d.

• Fig2b,e,h,c,f,I should have the same y axis units.

Response: The same y axis units are now used in Fig2b, c, e, f, h, i.

• The electron microscopy presented in Fig 4e does not show the classical 'beads on a string' phenotype, as that would involve multiple virions in a chain originating from a single budding site. What is shown here does appear to be a large number of virions that are nearly finished budding but remain attached/adjacent to the plasma membrane (ie, a budding defect)- but it is not strictly speaking "beads on a string".

Response: Unfortunately, we did not observe the “beads on a string” phenotype. We have revised the description in lines 187 to 190, “ Virions on the surface of MARCH8-overexpressing cells remain attached to the plasma membrane as opposed to the completely released IAV particles in the control cells, which suggests a virus budding defect phenotype associated with MARCH8 expression (Fig. 4e)”

• What about NP trafficking? Disrupting NP trafficking would also affect viral budding.

Response: We monitored NP trafficking at 6 hrs, 8 hrs and 10 hrs post-infection, and found that MARCH8 expression did not affect NP trafficking. The data are presented in Sup. fig. 1b.

• Line 202: decrease should be decreases.

Response: This error has now been corrected (line 203).

• It is difficult to see the M2 signal in Supp 4b,c, d when MARCH8 is expressed, which makes interpreting the suggested colocalization difficult . It would also be helpful to quantify this phenotype in more cells and provide a higher magnification/resolution image of the colocalization.

Response: We have now provided higher resolution images to better illustrate the colocalization phenotype. We have also calculated the colocalization pearson's coefficients and presented the data in Sup Fig 4c,d.

• Fig 5G, labelling MARCH8 as Flag, or Flag-M8 is confusing (I'm assuming a flag tagged MARCH8 is used as bait here?)

Response: The flag-tagged MARCH8 was used as bait. We have now corrected the labels in fig. 5G.

• There appears to be a 14 kDa ubiquitin modification on M2 that is not MARCH8 dependent (see band at 40 kDa in vector and M114A conditions in Fig 5i)- can the authors comment on how this may affect their conclusions?

Response: This weak 40kDa band in the vector and W114A samples is likely the result of the activity of background level of endogenous MARCH8, does not affect our conclusion that MARCH8 causes M2 ubiquitination.

• Line 243, ubiquitination is spelled wrong.

Response: This error has now been corrected (line 244).

• In the discussion (line 319), the correct references for influenza virus ESCRT independence are Bruce et al (pmid 19524996) and Chen et al (pmid 17475660).

Response: We have now cited Bruce et al (pmid 19524996) and Chen et al (pmid 17475660) for IAV independence of ESCRT in the discussion (line 320).

• The information on generation of mutant viruses is in two different sections in the methods (Cells, viruses and animals and Generation of mutant viruses) and this should be combined into a single section for clarity.

Response: We have now described the generation of mutant viruses in lines 443 to 456, as one section, “The M2 K78R mutant viruses mutPR8 and mutWSN were generated via the reverse genetic system as described previously. The M segment of PR8 or WSN virus with M2 K78R mutant was generated by site directed mutagenesis in the vRNA-mRNA bidirectional transcription vector pBD. Viruses were generated by transfecting HEK293T and MDCK co-cultured cells with the 8 plasmids. WSN, mutWSN, pdm09 H1N1 (A/Beijing/01/2009) and H3N2 (A/Beijing/30/95) viruses

were propagated in MDCK cells in minimum essential medium (MEM) supplemented with 0.5% bovine serum albumin (BSA) in the presence of 1 µg/ml tosylphenylalanyl chloromethyl ketone (TPCK)-treated trypsin, 1% (v/v) penicillin/streptomycin. Supernatants from the virus cultures were harvested three days post-infection. Virus stocks were aliquoted and stored at -80°C. Viral titers were calculated using plaque assays. The PR8 and mutPR8 viruses were inoculated into and propagated in 9-day-old specific-pathogen-free (SPF) chicken embryos for 3 days and then harvested and stored at -80°C until use.”

• The methods for the entry assays says attached but not internalized virions were removed by washing with PBS, the figure legend says they were removed with acid wash- which method was used?

Response: As suggested, we have now used exogenous NA to remove attached but not internalized virions. This method is now specified in the legend of fig.4a and 4b, “**a and b.** HEK293T cells were transfected with vector, MARCH8 or W114A. Cells were incubated with WSN virus (MOI = 5) at 4°C for 1 h (**a**) or then allowed to internalize bound IAV by incubation at 37°C for another 30 min before adding of exogenous NA to remove cell-surface virions. Viral binding or entry were assessed by determining the viral copy number in cell lysates by quantitative real-time PCR.”

• Please provide references for the reverse genetics system used.

Response: The reference Hoffmann, E., et al (pmid, 10801978) is now provided in the methods (line 444).

• What are the primer-probe sequences used for RT-PCR detection?

Response: The primer-probe sequences used for RT-PCR detection are now provided in the methods (line 4891-495).

• For electron microscopy were the cells fixed prior to scraping?

Response: Yes, cells were not fixed before scraping.

• If electron tomography rather than standard transmission electron microscopy was performed can the authors provide the 3D renderings/movies? Alternatively which type of microscopy was performed.

Response: The standard transmission electron microscopy was used to observe negative stained ultrathin sections. We clarified this in the methods (line 505-506).

Reviewer #5 (Remarks to the Author):

The reviewer thanks Guo and colleagues for satisfactorily revising the manuscript to address concerns raised by this reviewer. The revised manuscript requires careful proofreading before publication.

Response: We have proofread the manuscript.

Reviewer comments, second round –

Reviewer #4 (Remarks to the Author):

The authors have addressed my concerns, and the manuscript is suitable for publication.

Reviewer's comments

Reviewer #4 (Remarks to the Author):

The authors have addressed my concerns, and the manuscript is suitable for publication.

Response: Thank you for comments and careful reading of the manuscript.